**Title:** A data visualization tool for heritage buildings in India and Bangladesh: the project of a wiki for ID-SCAPES

**Authors:**
Giuseppe Resta*, 0000-0001-8489-5291
Sidh Losa Mendiratta*, 0000-0003-2960-8100
Tiago Trindade Cruz**, 0000-0001-8792-5142

*CEAU, Faculdade de Arquitectura, Universidade do Porto, Portugal
**CEAU, Faculdade de Arquitectura; CITCEM, Faculdade de Letras, Universidade do Porto, Portugal

**Themes:**
**Data visualisations and tools**. Visualizations and tools based on Wikidata and/or other Wikibase instances created or used by institutions, researchers and research projects.
**Projects and proposals**. Projects and proposals focusing on research and Wikidata and/or other Wikibase instances.

**Keywords**. Data visualization, heritage buildings, India, churches, preservation by record
**Format of the submission.** lightning talks
**Language of the presentation**. English and Italian (both if necessary)
**Confirmation of the physical presence in Florence of at least one of the authors**. Yes

**Abstract:**
In late 2023, the European Research Council approved the funding for the ID-SCAPES project to study and document the early modern religious architecture of the Christian minorities in India and Bangladesh. These religious sites are often multi-layered and contested heritage, and some buildings have suffered from increasing neglect, erasure and even effacement during recent decades. Furthermore, after the countries' independence, many within the Church hierarchies sought to distance themselves from colonial legacies, and many of the older churches were radically transformed or completely rebuilt with contemporary designs. This politically influenced process gained pace during the 1960s and continues today, evincing conflicting notions of heritage and identity.

Taking into consideration the risks arising from these processes and the progressive erasure of cultural heritage, ID-SCAPES aims to produce a Social History of the Built Environment of India and Bangladesh's medieval and early modern churches and sacral landscapes (built before ca. 1800), including both functioning and ruined buildings.

One of the principal outputs of the project will be herichurch.org, a wiki platform intended to provide a digital map, connected to the visual database, accessible to a wider audience.

Following the idea of "preservation by record" the project wikibase will combine 3D visualizations and CAD drawings as fundamental tools for cultural heritage conservation and research.

Challenging the European-centric historiographical framework that has been commonly employed for these themes, and through extensive fieldwork and the analysis of visual and written documents that remain unexplored, the project will advance a new methodological approach that embraces the buildings' complex histories. Addressing

issues such as caste, cultural "accommodation," "indigenous" agency, and local spatial and artistic traditions, ID-SCAPES will uncover the impact of such factors on church architecture.

Hence, the aim of ID-SCAPES' wiki is to visualize historical information gathering and processing data from research and fieldwork. In turn, such digital tools for endangered/contested sites can have an immediate impact on heritage management interventions.

For the Wikidata and Research 2025 conference, ideally in a "lightning talks" format, we will share our idea of how visual and written historical knowledge can be structured, possibly gathering suggestions from researchers who have achieved comparable outputs. As we are in the initial phase of the project, qualified feedback is of crucial importance to advance in the right direction.

The structure of knowledge should consist of a spatial database/digital map, visually organized for accessible consultation, and linked individual entries that constitute the main objects of research of ID-SCAPES. This system should work on a web mapping platform.

The authors have been involved in two comparable projects of heritage mapping, the hpip.org and eviterbo.fcsh.unl.pt platforms. In a similar vein, the Herichurch.org platform is expected to generate entries for a selection of about 50 representative sites during the project's timeline. Further content will be uploaded at later dates. This wiki platform is also one of the milestones of the project.

In summary, the ID-SCAPES wiki is set to advance the understanding and preservation of early modern religious architecture in India and Bangladesh through an innovative digital platform that captures and engages with the complex histories of these significant cultural sites.

## Authors Biography:

**Giuseppe Resta**. Giuseppe Resta is an integrated researcher at the Faculdade de Arquitectura da Universidade do Porto (Centro de Estudos de Arquitectura e Urbanismo). He previously held teaching positions at Yeditepe University (Istanbul) and Bilkent University (Ankara) as an assistant professor, at Politecnico di Bari (Bari) as an adjunct professor, and at Polis University (Tirana) as a lecturer. Resta received his Ph.D. in architecture from Università degli Studi RomaTRE (Roma, with fellowship, Doctor Europaeus label, summa cum laude) and his M.Arch from Politecnico di Bari (Bari, summa cum laude). He is the owner and curator of Antilia Gallery (IT) and co-founder of the architecture think tank PROFFERLO Architecture (IT-UK). His work has been published in architectural journals such as The Plan Journal, STUDIO, FAM, Architecture and Urban Planning, CIDADES. His latest monographic books are "Journey to Albania: Architectures, expeditions and landscapes of tourism" (Accademia University Press, 2022) and "The City and the Myth" (Libria, 2023). Resta co-led the WG2 COST Action CA18126 "Writing Urban Places".

**Sidh Losa Mendiratta**. Sidh Losa Mendiratta is an Assistant Professor at the Faculty of Architecture of the University of Porto, and an Integrated Researcher at the Centre for Studies in Architecture and Urbanism. Since September 2024 he is the Coordinator of the research project "ID-SCAPES: Building Identity. Religious Architecture and Sacral Landscapes of Christian Minorities in India and Bangladesh (ERC/COG/101125057)" funded by a Consolidator Grant of the European Research Council (2024-2029), and since January 2022 he is the Co-Principal Investigator of the research project

"PORTofCALL. African-Asian-European Encounters: Cultural Heritage and Ports of Call in the Indian Ocean during the Early Modern Period (FCT/PTDC/ART-DAQ/4357/2021)" funded by FCT (2022-2025). He holds a PhD in Architecture from the University of Coimbra (2012, summa cum laude) funded by an FCT fellowship. Between 2012 and August 2024, he was an Integrated Research at the Centre for Social Studies of the University of Coimbra, and an Assistant Professor at Lusófona University of Porto, holding the chair of History of Portuguese Architecture. In 2013 he was awarded the "Fernando Távora" prize by the Order of Architects of Portugal. He is the author of the books "The Church and Convent of Our Lady of Mount Carmel, Chimbel" (2021) and "Domus-fortis in Æquator: a segunda vida da casa-torre de origem Europeia no antigo Estado da Índia" (2019). In 2011, he was a visiting scholar at the Xavier Centre for Historical Research, Goa. Specializing in cultural heritage of Portuguese influence in South and Southeast Asia, he has conducted twenty-five georeferenced topographic surveys of archaeological and architectural sites in India, collaborating with the Archaeological Survey of India, Fundação Oriente and Fundação Calouste Gulbenkian. He is author and/or coauthor of over seventy papers, including five papers presented at the Society of Architectural Historians International Conference (2008, 2009, 2010, 2018 and 2024).

**Tiago Trindade Cruz**. Tiago Trindade Cruz (1985) holds a PhD in Heritage Studies from FLUP (2022), a Master's in Virtual Heritage from the University of Alicante (2020), and an Integrated Master's in Architecture from FAUP (2010). He is currently a Junior Researcher on the project ID-SCAPES - Building Identity: Religious Architecture and Sacral Landscapes of Christian Minorities in India and Bangladesh. Since the 2021-22 academic year, he has been a Visiting Assistant Professor at FLUP, where he teaches History of Contemporary Architecture. He has been an integrated researcher at CITCEM/FLUP and a collaborator at CEAU/FAUP since 2023. Tiago is also part of the UNESCO Chair "Heritage, Cities and Landscapes" hosted by FAUP. Between 2022 and 2024, he collaborated on the Nomination of Álvaro Siza's Works of Architecture for inscription on the World Heritage List, during which he held a scholarship as a Post-Doctoral Fellow.

**Authors bibliography:**

- Resta, Giuseppe, "Mapping the Invisible: Problems of Interpretation and Representation of Hormuz Island, Iran," *Athens Journal of Architecture* 11, 1 (2025): 1-32
- Resta, Giuseppe, & Dicuonzo, Fabiana (2024), "Towards a Digital Shift in Museum Visiting Experience. Drafting the Research Agenda Between Academic Research and Practice of Museum Management." In M. Barberio, M. Colella, A. Figliola, & A. Battisti (Eds.), *Architecture and Design for Industry 4.0: Theory and Practice* (2024): 609-648. Cham: Springer.
- Resta, Giuseppe; Dicuonzo, Fabiana; Karacan, Evrim; & Pastore, Domenico, "The impact of virtual tours on museum exhibitions after the onset of covid-19 restrictions: visitor engagement and long-term perspectives." *SCIRES-IT* 11,1 (2021): 151-166.
- Mendiratta, Sidh Losa, "An Early Modern Sacro Monte in Mumbai," Journal of the *Society of Architectural Historians* 82, 2 (June 2023): 150-169.
- Mendiratta, Sidh Losa, Fernando Dias Velho, *The Church and Convent of Our Lady of Mount Carmel, Chimbel*, Panjim: Fundação Oriente / MCCAC, 2021.

- Mendiratta, Sidh Losa, *Domus-fortis in Æquator: A segunda vida da casa-torre de origem Europeia no antigo Estado da Índia,* Porto: Ordem dos Arquitectos - Secção Regional Norte, 2019.

- Cruz, Tiago Trindade; Cunha Ferreira, Teresa; Pedro Murilo Freitas Gonçalves; David Ordóñez Castañón; Vasconcelos, Domingas. "Álvaro Siza's Lessons on Contextual Design in Historic Areas: Projects for the Historic Centre of Porto (1968-2000)." *Conservation / Sustainable Design. Heritage Challenges in Historic Urban Landscapes. EAAE*, 287–307.

- Cruz, Tiago Trindade; Santos, Marisa Pereira. "Cultural Heritage and Mediation: The Use of ICT in the Communication of the Artistic Layers of the Church of Saint John the Baptist of Foz do Douro." *Mimesis Journal* (forthcoming).

- Ferreira, Teresa Cunha; Cruz, Tiago Trindade; Freitas, Pedro Murilo; Genin, Soraya Monteiro. "Methodology for the Digital Documentation of Modern Architecture: Applied Research on Álvaro Siza's Works for the World Heritage List." *The International Archives of the Photogrammetry, Remote Sensing and Spatial Information Sciences*, XLVIII-M-2, 541-547.
