# OpenReview forum: "A data visualization tool for heritage buildings in India and Bangladesh: the project of a wiki for ID-SCAPES"
_wikimedia.it/Wikidata_and_Research/2025/Conference — WD&R LT_

### Official Review · ~Elena_Marangoni1 · 2025-01-05
**Architecture and history in structured data and geographical data**

**Originality:** 5
**Impact:** 5
**Confidence:** 4

**Review:**

This project has some point in common with the submission #45 "eViterbo", as declared by the authors, but is at the first steps, so they proposes the lightning talk format that can be useful for a comparison and exchange of ideas and projects with other researchers. It is very interesting and complex, aiming at putting together architecture and cultural heritage preservation  along with history, and in particolar colonial history.

**Compliance:**

5

**Scientific Quality:**

5

---

### Official Review · ~Iolanda_Pensa1 · 2025-01-12
**Wikibase but potentially also Wikidata and other Wikimedia projects**

**Originality:** 4
**Impact:** 5
**Confidence:** 4

**Review:**

I find the topic fascinating, and it is relevant not only for Wikibase but also for Wikidata directly.

The topic of heritage is always very relevant for the Wikimedia projects and the fact that India and Bangladesh have freedom of panorama can facilitate not only the collaboration with Wikidata but also the upload of images and other content on Wikimedia Commons (and those uploads are easily possible thanks to the use of open licenses such as CC BY 4.0 always requested by the open science policy of the European projects).

Furthermore, more content about architecture in India and Bangladesh can contribute to filling Wikidata and Wikimedia knowledge gaps. There are also very active Wikimedia communities in both countries (maybe interesting to involve).

A very interesting lightening talk.

**Compliance:**

5

**Scientific Quality:**

5

---

### Decision · Program_Chairs · 2025-01-23

**Decision:**

Accept (LT)

**Comment:**

== presence confirmed==